# Post-COVID-19 Vaccine Hyperproduction of Anti-Spike Antibodies and Rheumatological Manifestations

**DOI:** 10.3390/vaccines13101028

**Published:** 2025-09-30

**Authors:** Marc Alexandre Golstein

**Affiliations:** Department of Rheumatology, Saint-Jean Hospital, Brussels 32 Boulevard du Jardin Botanique, 1000 Brussels, Belgium; marc.golstein.pcr@gmail.com; Tel.: +32-2-221-9987; Fax: +32-221-9828

**Keywords:** COVID-19 vaccines, arthritis, polymyalgia, spike protein

## Abstract

Introduction: Vaccines are the most widely used public health measure to control the global COVID-19 pandemic. Most vaccines used in Europe and North America are mRNA-based. A mass vaccination campaign was carried out between 2021 and 2024. Some adverse events have been reported based on analogies with previous virus-attenuated vaccines. Objectives: Given the new mechanism inducing specific antibodies, we questioned the role of mRNA Spike vaccines and the significance of hyperproduction of anti-Spike antibodies in the emergence of early and late onset rheumatological manifestations observed after one or more injections. Material and Methods: A prospective observational study involving two cohorts was initiated. The first cohort was observed from 13 September 2021 to 30 September 2022, and the second cohort from 1 October 2022 to 30 September 2023. The study also focused on the interval between the last vaccine injection and the onset of rheumatic symptoms. None of the patients had a history of rheumatic or inflammatory diseases. We compared both cohorts and ankle arthritis case series to analyze the differences between early and late-onset adverse events. Results: In both cohorts and case series, the majority of patients were women. The most common symptoms included diffuse muscle pain, which mimics polymyalgia rheumatica, and ankle arthritis. Very high levels of anti-Spike antibodies (>2080 BAU/mL) were generally detected. The Pearson correlation coefficient between both cohorts and case series was very high, confirming the reproducibility of post-vaccine clinical and biological features. Conclusions: These rheumatological manifestations might be triggered by inappropriate individual immune responses to the vaccine’s Spike protein and/or the overproduction of Spike protein, which can mediate a pro-inflammatory reaction, explaining early and late-onset effects.

## 1. Introduction

Vaccination is the most widely used public health measure to control the global COVID-19 pandemic. Most vaccines used in Europe and North America are mRNA-based, such as BNT162b2 and mRNA-1273, but viral vector DNA vaccines like ChadOx1 and CoV-19 are also used [1,2]. A mass vaccination campaign was launched between 2021 and 2024. Some adverse events have been reported, drawing analogies from side effects induced by attenuated or inactivated virus vaccines. A few rheumatological events have been documented [3,4], occurring rapidly after the administration of one or more doses. The Spike protein targeted by the humoral immune response binds to the ACE2 receptor, blocking SARS-CoV-2 entry and preventing inflammation. However, the Spike protein also binds to the alpha estrogenic receptor and the Toll 2 and 4 receptors [5,6]. Furthermore, the feedback mechanisms governing the quantity and quality of foreign protein production by the endoplasmic reticulum and Golgi apparatus in humans remain unknown [7]. Due to the different and specific mechanisms of Spike protein RNA vaccines or viral vector DNA vaccines, we question the role of COVID-19 vaccines in the emergence of early- and late-onset rheumatological manifestations observed following one or more injections.

## 2. Material and Methods

This clinical study was approved by Saint-Jean hospital’s ethical board in accordance with the Belgian Law regarding the use of medical data, which is available for any additional request. All patients were examined and followed in the same rheumatology department. A prospective observational study of 2 cohorts was launched from 13 September 2021 to 30 September 2022 for the first one and from 1 October 2022 to 30 September 2023 for the second one. After the first year, we focused the observational study also on the delay between the last vaccine injection and the beginning of rheumatic symptoms. We exclude all patients from the study if they have a past of rheumatic, inflammatory, granulomatous, or autoimmune diseases. As we also excluded from our study anyone with anxiety, depression, fibromyalgia, or any other psychiatric condition, we did not prospectively study the HPA ACTH cortisol axis. Furthermore, none of the patients included presented with complaints or clinical symptoms related to abnormal ACTH-cortisol activity. Specific tests such as CH50, C3, C4, ACE, or immunoglobulin typing were not systematically requested at the start of the study. These were performed on an exceptional basis depending on clinical and/or biological findings. No abnormalities were found except for elevated ACE in the case of sarcoidosis (excluded of the study) that appeared 3 months after RNA vaccination. In a series of 184 patients presenting with musculoskeletal symptoms joint and skeletal symptoms after COVID-19 vaccination, we report an initial cohort of 124 patients with no interfering diseases who developed rheumatological manifestations after vaccination with an mRNA vaccine encoding the SARS-CoV-2 spike protein (120/124) or a viral vector DNA vaccine (4/124). Of 156 patients, 31 were excluded according to the same criteria, and a second cohort of 125 patients who developed late adverse effects was studied. None of the patients had a history of rheumatological disease or developed rheumatological disease during the follow-up. All causes of rheumatic diseases were excluded by a full biological examination and adequate imaging according to the clinic. We also focused our biological investigation on anti-nuclear antibodies (ANA), rheumatoid factor (RF), anti-citrullinated peptide (anti-CCP), and anti-neutrophil cytoplasmic antibodies (ANCAs). A post-vaccination blood serological follow-up for anti-Spike antibodies was requested several times for all patients at diagnosis and during the follow-up. None of the subjects had a suspicion of viral infection as confirmed by the diverse samplings collected. All the patients were treated with a low oral dose of 8 mg/day of methylprednisolone, reduced to 4 mg/day after 2 weeks. We double-checked the medical history of each patient (both cohorts) by consulting ABRUMET (professional access to general patient medical data base) according to the Belgian law. None of the patients developed any other disease after at least 24 months of follow-up. All patients underwent serological testing for SARS-CoV-2 after vaccination. IgG antibodies anti-Spike to SARS-CoV-2 were determined with LIAISON XL (DiaSorin S.p.A., Saluggia, Italy) using the IgG SARS-CoV-2 Trimerics reagent (DiaSorin S.p.A.). The measurement range was between 4.81 and 2080 BAU/mL, with a threshold of 33.8 BAU/mL. According to the company, a value of 520 BAU/mL or more corresponds to a high neutralization capacity of the antibodies. As this serological test is not reimbursed by the Belgian social security, each patient had to pay for it. Because of ethical board restriction, we were only allowed to use the serological results of our own vaccinated patients as control group followed at the consultation with other diagnoses. A vaccinated control group of 59 subjects (mean age 56 years old, 43 females, and 16 males) did not reveal such a high level of anti-Spike antibodies in the blood sample. The mean anti-Spike antibody level in the control group was 486.7 BAU/mL (4.8–842). We also tested 6 unvaccinated patients of our own consultation to assess the control group (mean value 54 BAU/mL). We questioned each patient reporting adverse events. We used the Pearson test to compare early and late onset cohorts and bilateral ankle arthritis after vaccination to examine the potential link between COVID-19 vaccines. We also studied vaccinated patients with sarcoidosis during the observational period.

## 3. Results

### 3.1. Cohort 1

Among the 124 patients (median age 51.6 years [17–86 years]), 37 are men (33 Caucasians, 3 Africans, 1 Asian) and 87 (70%) are women (median age 54.6 years [20–80 years]), including 79 Caucasians, 7 Africans, and 1 Asian. (Table 1) Of these patients, 73 (58.9%) had polymyalgia, 38 (30.6%) had bilateral ankle arthritis, 1 had monoarthritic of the ankle, 13 (10.4%) had arthritis of the fingers, hand, or wrist, 9 (7%) had arthritis of both knees, 1 had monoarthritic of the knee, 2 had arthritis of the hips, and 1 had seronegative polyarthritis. A combination of rheumatological symptoms occurred in 19 patients (Table 2, Figure 1). Antinuclear antibodies (ANA) were detected in three patients, and anti-citrullinated peptide antibodies (anti-CCP) were detected in two patients (Table 3, Figure 2). ANCAs were not detected in any patients and in the absence of signs of vasculitis or autoimmune disease, the complement pathway was not investigated. Most symptoms appeared after the 2nd or 3rd dose of the vaccine: 17 cases on average 12.5 days (range 1–42 days) after the first dose, 50 cases on average 20.3 days (range 1–90 days) after the second dose, 53 cases on average 30.5 days (range 1–90 days) after the third dose, and 4 cases on average 30.5 days (range 1–90 days) after the fourth dose. Very high levels of anti-Spike antibodies (>2080 BAU/mL) were found in 104 cases (84%), and high levels (average 1318 BAU/mL) were found in 20 cases (16%). In some cases, the anti-Spike antibody level increased during follow-up in the absence of a proven COVID-19 infection or vaccine booster and correlated with an outbreak of muscle or joint pain. At the beginning of the follow-up, a biological inflammatory syndrome characterized by elevated sedimentation rate (SR) and/or C-reactive protein (CRP) was found in 40 out of 124 patients (32%), especially in women (32 out of 40, 80%). The mean SR/CRP values were 41 mm/h/14.6 mg/L in women and 46.5 mm/h/21.3 mg/L in men. Even if most patients had no inflammatory abnormalities, they were symptomatic and responded to a low dose of 8 mg methylprednisolone per day, which was rapidly reduced to 4 mg per day.

### 3.2. Cohort 2

Among the 125 patients (median age 50.6 years [28–85 years]), 31 are men (29 Caucasians and 2 Asians) and 94 (75%) are women (median age 41.2 years [17–85 years]), including 86 Caucasians, 7 Africans, and 1 Asian (Table 1). Of these patients, 53 (42.4%) had polymyalgia, 37 (29.6%) had bilateral ankle arthritis, 33 (26.4%) had arthralgia or arthritis of the fingers, hand, or wrist, 15 (12%) had arthritis of both knees, 9 (7.2%) had monoarthritic of the knee, 3 had arthritis of the hips, and 1 had seronegative polyarthritis. A combination of these rheumatological symptoms occurred in 24 patients (Table 2, Figure 1). Anti-nuclear antibodies (ANA) were detected in 8 patients, rheumatoid factor in 1 patient, and anti-citrullinated peptide antibodies (anti-CCP) in 3 patients (2 slightly positive and 1 with high levels) (Table 3, Figure 2). As ANCAs were not detected in the first cohort, we did not measure them again. No abnormalities in CH50 C3 and C4 testing were found in a patient presenting with livedo reticularis. Most symptoms appeared after the 2nd or 3rd dose of the vaccine: 4 cases on average 12.5 days after the first dose, 45 cases on average at least 122 days after the second dose, 58 cases on average at least 113 days after the third dose, and 18 cases on average at least 126 days after the fourth dose. Very high levels of anti-Spike antibodies (>2080 BAU/mL) were found in 96 cases (76.8%), and high levels (average 1435 BAU/mL) were found in 29 cases (23.2%). A biological inflammatory syndrome characterized by elevated sedimentation rate (SR) and/or C-reactive protein (CRP) was found in 46 out of 125 patients (36.8%), with 35 out of 49 women (71.4%) affected. The mean SR/CRP values were 19 mm/h/7.04 mg/L in women and 15.7 mm/h/12.2 mg/L in men. Most patients in the second cohort had no inflammatory abnormalities; they were symptomatic and responded equally to a low dose of 8 mg methylprednisolone per day, which was rapidly reduced to 4 mg per day (the same treatment and follow-up as for the first group). In both cohorts, none of the patients reported any adverse events related to COVID-19 vaccines due to fear and lack of knowledge about the reporting process. No serum sickness related to high levels of anti-Spike antibodies was observed. The Pearson correlation coefficient between both cohorts is very high (0.95), confirming the reproducibility of clinical and biological features.

**Table 1 vaccines-13-01028-t001:** Cohort description.

	Patients	Age		Gender		Ethnic			Vaccines		Vaccine Doses/Delay Days	
		F	M	F	M	Caucasian	Black African	Asian	mRNA	DNA	1st/Delay	2nd/Delay
											3rd/Delay	4th/Delay
Cohort 1	124	54.6	59.6	87	37	79 F/32 M	7 F/3 M	1 F/1 M	120	4	17/12, 5 d	50/20, 3 d
											55/30, 5 d	4/30, 5 d
Cohort 2	125	41.2	50.6	94	31	86 F/29 M	7 F/0 M	1 F/2 M	123	2	4/12, 5 d	45/>122 d
											58/>113 d	18/>126 d

**Figure 1 vaccines-13-01028-f001:**
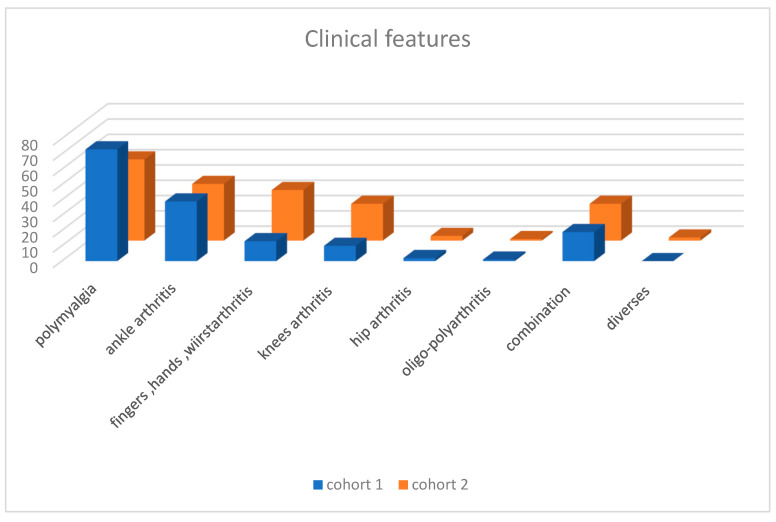
Ankle arthritis; 38 bilateral + 1 unilateral; Fingers, hand and wrist: arthritis and arthralgia; knees arthritis: 9 and 15 bilateral, 1 and 9 unilateral arthritis.

**Table 2 vaccines-13-01028-t002:** Clinical features.

	Polymyalgia	Ankle Arthritis	Fingers Hands Wrist	Knees Arthritis	Hips Arthritis	Ploy—Oligo Arthritis	Combination	Diverse
Cohort 1	73	38 + 1	13	9 + 1	2	1	19	0
Cohort 2	53	37	33	15 + 9	3	1	24	2

Ankle arthritis: 38 bilateral + 1 unilateral; fingers, hand and wrist; arthritis and arthralgia; knees arthritis: 9 and 15 bilateral, 1 and 9 unilateral arthritis.

**Figure 2 vaccines-13-01028-f002:**
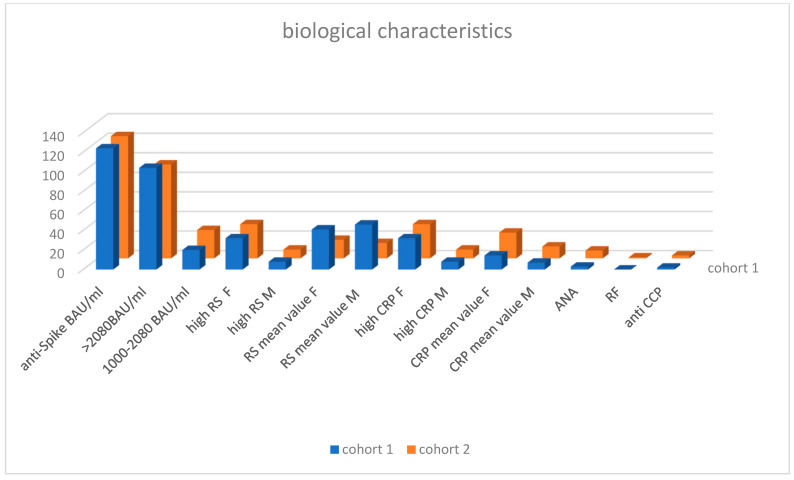
Anti Spike antibodies: plasmatic level in BAU/mL, high RS or CRP: cohort 1 40 pateitns cohort 2 46 patients, ANA antinuclear antibodies, RF rheumatoid factor, anti CCP anti citrullinated peptide.

**Table 3 vaccines-13-01028-t003:** Biological characteristics.

	Anti-Spike BAU/mL		High SR mm/h Patient/Mean Value		High CRP mg/L Patient/Mean Value		ANA	RF	Anti CCP
	>2080	1000–2080							
Cohort 1	104	20/1318	32 F/41 mm/h	8 M/46 mm/h	32 F/14.6 mg/L	8 M/26.3 mg/L	3	0	2
Cohort 2	96	29/1434	35 F/19 mm/h	9 M/15.7 mm/h	35 F/7.04 mg/L	9 M/12.2 mg/L	8	1	3

Anti-Spike antibodies: plasmatic level in BAU/mL, high RS or CRP; cohort 1—40 patients, cohort 2—46 patients, ANA: antinuclear antibodies, RF: rheumatoid factor, anti CCP: anti citrullinated peptide.

Comparison between early and late onset bilateral ankle arthritis (Table 4 and Table 5, Figure 3)

We compare the first 17 patients of the cohort having late-onset bilateral ankle arthritis with our initial case series. Among the 17 patients (median age 57.1 years [38–85 years]), 14 are women, with 10 identifying as Caucasian, 3 as Black African, and 1 as Asian. The three men are all Caucasian. Three women developed bilateral ankle arthritis six months or more after receiving two doses of mRNA vaccines, one after two doses of DNA vaccines, twelve (ten women and two men) after three doses, and one man after four doses. Only three patients had received a viral vector DNA vaccine before the mRNA boosters (one or two doses). The late-onset case series is compared with our initial published series of early post-vaccine ankle arthritis. Clinical and biological features between early and long-latency patients are summarized in Table 4. High levels of anti-Spike antibodies were detected in blood samples at the time of diagnosis, and these elevated levels persisted in 9 of the 17 patients studied for more than 11 months after the diagnosis of ankle arthritis. Among these cases, four experienced a relapse of bilateral ankle arthritis. The number of vaccine doses is higher in the late-onset series (47) than in the early-onset group (25). The number of DNA and RNA vaccine doses received is also higher in the late-onset group (eight DNA/39 RNA doses) than in the early-onset case series (two DNA/23 RNA doses). We also focused on the delay between the last vaccine dose and the rheumatological diagnosis. The mean delay is 17 months in the late-onset group and 0.44 months in the early-onset case series. Excluding the delay parameter, the Pearson correlation coefficient (0.91) between both case series confirms clinical reproducibility.

Flare and new onset of sarcoidosis (Table 6)

Patients experiencing a new onset or flare of sarcoidosis after receiving COVID-19 vaccines also developed bilateral ankle arthritis. Three patients developed symptoms shortly after the second dose of the COVID-19 vaccine (one DNA and two mRNA), while two others had a delayed onset (both with mRNA vaccines). The sixth patient could not specify the delay between the last vaccine and the onset of the articular flare. All individuals exhibited elevated levels of anti-Spike protein, with women constituting most participants (five out of six). The clinical evolution was favorable under treatment with a low dose of prednisolone.

**Figure 3 vaccines-13-01028-f003:**
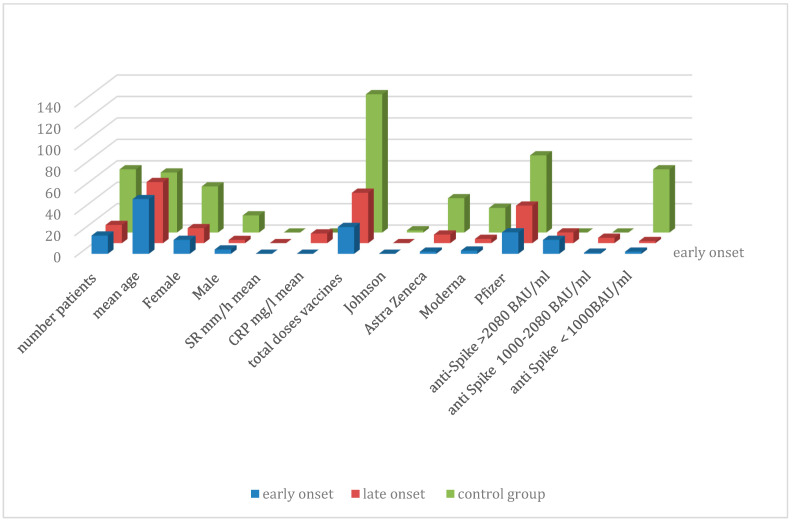
Comparison between early and late onset ankle arthritis.

**Table 4 vaccines-13-01028-t004:** Comparison control group, early-onset, and late-onset case series.

Patients			VS/CRP		Number Vaccine Doses				Anti-Spike Antibodies BAU/mL		
	Age	Gender	VS mm/h	CRP mg/L	Johnson	Astra Zeneca	Moderna	Pfizer	>2080	1000–2080	<1000
59 Control	56	43 F/16 M			2	32	23	72	0	0	59
17 Early	51.8	13 F/4 M	27.7 (3–85)	23.7 (2–69)	0	2	3	20	13	1	2
17 Late	57.1	14 F/3 M	26.6 (2–73)	9 (2–52)	0	8	4	35	10	5	2

**Table 5 vaccines-13-01028-t005:** Clinical characteristics of the late-onset arthritis patients.

Case	Sex/Age	Ethnic	Vaccine	Clinical Features	ESR (mm/h)/CRP (mg/L)	SarsCov2 S IGG AB (Delay)
1	F/48	Caucasian	Pfizer × 2	Ankle R + L	73/4	>2080 (17 months)
2	F/75	Caucasian	Pfizer × 3	Ankle R + L	48/9	>2080 (19 months)
3	M/60	Caucasian	Pfizer × 3	Ankle R + L	7/2	1300 (22 months)
4	F/52	Caucasian	Pfizer × 3	Ankle R + L	22/4	>2080 (13 months)
5	F/53	Caucasian	Pfizer × 3	Ankle R + L	15/2	425 (15 months)
6	M/50	Caucasian	Pfizer × 3	Ankle R + L	26/21	>2080 (13 months)
7	F/80	Caucasian	Moderna × 3	Ankle R + L	30/04	>2080 (16 months)
8	F/74	Caucasian	Pfizer × 3	Ankle R + L	11/52	>2080 (15 months)
9	F/54	Caucasian	AZ × 2	Ankle R + L	39/15	1280 (22 months)
10	F/68	Caucasian	AZ × 2 Pfizer × 1	Ankle R + L Knee R	21/02	1680 (17 months)
11	F/55	Caucasian	Pfizer × 2	Ankle R + L	39/4	>2080 (25 months)
12	M/69	Caucasian	AZ × 2 Pfizer × 2	Ankle R + L	6/2	641 (8 months)
13	F/39	Asian	Pfizer × 2 Moderna × 1	Ankle R + L	30/06	>2080 (17 months)
14	F/48	African	Pfizer × 2	Ankle R + L	56/8	1530 (20 months)
15	F/38	African	Pfizer × 3	Ankle R + L	19/10	1550 (11 months)
16	F/49	African	AZ × 2 Pfizer × 1	Ankle R + L	2/6	>2080 (19 months)
17	F/57	Caucasian	Pfizerx2Moderna × 1	Ankle R + L Knee R + L	9/3	>2080 (21 months)

**Table 6 vaccines-13-01028-t006:** Flare and new onset of sarcoidosis.

	Age	Gender	Ethnic	Biology		Anti-Spike BAU/mL	Clinical	Vaccine	Delay
				SR mm/h	CRP mg/L				
1	27	F	Black African	27	16	>2080	Ankle L + R	Moderna × 2	7 days
2	66	F	Black African	44	17	1780	Ankle L + R	AZ × 2	5 days
3	38	F	Black African	6	4	>2080	Ankle L + R	Pfizer × 2	7 days
4	33	F	Maghreban	50	24	>2080	Ankle L + R and mediastinal lymphadenopathy	Pfizer × 2	3 months
5	43	M	Black African	32	24	1250	Ankle L + R	Moderna × 2	>6 months
6	48	F	Black African	20	2	>2080	Ankle L + R	Pfizer × 2	Unknown

## 4. Example of Longitudinal Case Studies

Case 1: A 71-year-old Caucasian woman with a medical history of hypertension and hypercholesterolemia consulted for bilateral ankle synovitis that appeared 8 months ago, 15 days after receiving the third dose of the Pfizer vaccine. At the time of consultation, blood tests revealed a CRP level of 2 mg/L, no autoantibodies, and anti-Spike antibodies > 2080 BAU/mL (8 months post-vaccine). Comprehensive investigations for bilateral arthritis showed no abnormalities. The patient improved with 8 mg/day of methylprednisolone and was instructed to reduce the dose to 4 mg/day after 15 days. After 1 month, a follow-up blood test showed an SR of 4 mm/h, CRP of 2 mg/L, and anti-Spike antibodies > 2080 BAU/mL (9 months post-vaccine). The synovitis had disappeared, and the methylprednisolone dose was reduced to 2 mg/day. Four months later, at a follow-up consultation, the SR was 10 mm/h, CRP was 1 mg/L, and anti-Spike antibodies remained >2080 BAU/mL (13 months post-vaccine). The patient requested to stop the corticosteroid treatment. Three months later, the patient consulted again due to the reappearance of bilateral ankle synovitis. Usual tests did not show elevated SR (6 mm/h) or CRP (2 mg/L), but anti-Spike antibodies remained >2080 BAU/mL (16 months post-vaccine). The patient improved with 4 mg of methylprednisolone. Once again, she stopped the treatment and later consulted for bilateral ankle arthritis. Blood test results showed an SR of 4 mm/h, CRP of 20 mg/L, and anti-Spike antibodies > 2080 BAU/mL (20 months post-vaccine). The patient recovered rapidly with a daily dose of 4 mg of methylprednisolone. This time, she maintained the treatment, and follow-up tests showed normal SR (4 mm/h) and CRP (1 mg/L), but anti-Spike antibodies remained very high (>2080 BAU/mL 25 months post-vaccine).

Case 2: A 56-year-old Caucasian woman with no medical history consulted for bilateral shoulder pain and polymyalgia that appeared 17 months ago, 2 months after receiving the second dose of the Pfizer vaccine. At the time of consultation, blood tests revealed an SR level of 26 mm/h, a CRP level of 27 mg/L, no autoantibodies, and anti-Spike antibodies at 1070 BAU/mL (17 months post-vaccine). Investigations for bilateral arthritis showed no abnormalities. The patient improved with 8 mg/day of methylprednisolone and was instructed to reduce the dose to 4 mg/day after 15 days. She stopped the treatment spontaneously and later consulted for polymyalgia. Blood test results showed an SR of 38 mm/h, CRP of 32 mg/L, and anti-Spike antibodies > 2080 BAU/mL (22 months post-vaccine). The patient recovered rapidly with a daily dose of 4 mg of methylprednisolone. She maintained the treatment, and follow-up tests showed an SR of 29 mm/h, CRP of 33 mg/L, and anti-Spike antibodies remained high at 1160 BAU/mL (31 months post-vaccine).

Case 3: A 55-year-old Caucasian man with a medical history of cervical osteoarthritis consulted for bilateral ankle synovitis and polymyalgia that appeared 8 months ago, more than 3 months after receiving the fourth dose of the Pfizer vaccine. At the time of consultation, blood tests revealed an ESR of 33 mm/h, a CRP level of 27 mg/L, no autoantibodies, and anti-Spike antibodies > 2080 BAU/mL (8 months after the last vaccine). Tests for bilateral arthritis revealed no abnormalities. The patient’s condition improved with 8 mg/day of methylprednisolone and was instructed to reduce the dose to 4 mg/day after 15 days. Blood test results showed an ESR of 2 mm/h, CRP of 2 mg/L, and anti-Spike antibodies > 2030 BAU/mL (13 months after vaccination). He continued treatment, and follow-up tests showed an ESR of 2 mm/h, a CRP of 1 mg/L, and anti-Spike antibodies still elevated at 1270 BAU/mL (17 months after vaccination). Corticosteroid treatment was discontinued.

## 5. Discussion

The limitations of this study stem from its nature as an observational clinical study conducted at a single rheumatology center and the lack of sufficient reference material on the subject in 2021. No information was or is available regarding the persistence and significance of anti-Spike protein antibodies more than six months after the last vaccination. During the prospective follow-up, we observed the persistence of very high levels of anti-Spike protein antibodies, which is currently our only follow-up marker, allowing us to compare the two patient cohorts. The consistent attendance of patients at the department over the two years (124 patients in the first year and 125 in the second) of observational follow-up provides strong clinical evidence, enabling us to observe the presence of delayed rheumatological side effects independent of rheumatological history. A few cases of reactive arthritis following COVID-19 infection have been reported. The incidence appears to be very low and inversely proportional to the severity of the disease [8]. As only vaccinated patients were included in the study, the vast majority of whom had received two or three doses, the risk of post-infectious rheumatological or inflammatory manifestations is very limited. The choice of anti-Spike antibody dosage was motivated by the mode of action of anti-COVID-19 vaccines, which focus on the Spike protein [1,2]. On the other hand, during coronavirus infections, anti-Spike antibodies have an antibody-dependent enhancement (ADE) effect, promoting viral infection and stimulating certain pro-inflammatory cytokines [9,10]. This also raises questions about the possible role of hyperproduction of anti-Spike protein antibodies in the emergence of rheumatological manifestations by fixing viral particles (Spike protein) and mimicking the ADE mechanism. Unlike all other viral serology tests, anti-COVID-19 serology tests are not covered by the social security in Belgium, which has limited access to other tests such as anti-N tests. We also chose to exclude patients with anxiety, depression, fibromyalgia, chronic fatigue syndrome, or other psychiatric conditions to avoid any diagnosis of long COVID [11], whose symptoms are essentially subjective and overlap with those of other conditions. In both cohorts and case series, a large majority of patients are women with an average age of 50.6 and 51.6 years. In our study, women are more likely than men to experience a biological inflammatory syndrome. Estrogens have a favorable anti-inflammatory and immunomodulatory effect, providing protection during the SARS-CoV-2 epidemic [12]. The selective affinity of the Spike protein for estrogen alpha receptors [5] could also explain the predominance of biological inflammatory syndrome in women, as it competes with endogenous or replacement estradiol for binding to its receptor during menopause. The significance of gender influence encourages us to explore the subject further. The most common complaint is diffuse muscle pain that resembles polymyalgia rheumatica in presentation [13,14], typically occurring in individuals in their early 50s and often without an inflammatory syndrome. Given the specific joint and muscle symptoms and the potential pro-inflammatory mechanisms mediated by either the Spike protein or anti-Spike antibodies, a low daily dose of corticosteroids was administered [15]. After excluding most causes of diffuse pain, 8 mg of methylprednisolone per day, reduced to 4 mg per day, was administered for other clinical manifestations and proved effective in most cases (65 out of 73 in cohort 1 and 35 out of 53 in cohort 2). The most frequent joint involvement is bilateral arthritis of the ankles, with 38 cases (plus 1 unilateral ankle arthritis) in cohort 1 (31%) and 37 cases in cohort 2 (29.6%). In our initial study of 17 cases of early-onset bilateral ankle arthritis [16], 16 out of 17 patients responded favorably to low-dose corticosteroid therapy, and 13 out of 17 in the series of late-onset ankle arthritis. New cases of inflammatory mono/oligo arthritis involving the ankle are more likely to be due to reactive undifferentiated arthritis or spondylarthritis than rheumatoid arthritis and microcrystalline arthritis [17]. Abhishek and colleagues [18] reported that in the Birmingham early inflammatory arthritis (BEACON) cohort, patients with bilateral ankle synovitis were more likely to be classified as having acute sarcoid arthritis. We remind that all those diagnoses were excluded from both cohorts. About 75 cases of acute sarcoidosis are expected to be diagnosed in one year for the Brussels region and shared into 13 hospitals. The 6 patients (excluded from both cohorts) having a previous articular sarcoidosis and those who developed a flare or sarcoidosis after vaccination [19,20,21] confirm the normal incidence of the disease diagnosed in our department and raise questions about the significance of the high level of anti-Spike antibodies and the role of Spike protein in the occurrence of inflammatory articular manifestations. We find a total of 46 cases (13 early onset cohort + 33 late onset cohort) of arthralgia, arthritic fingers, and wrists without any erosions. No other cause was discovered during the follow-up for patients with other articular locations (knees or hips). No patients exhibited serum sickness symptoms associated with elevated anti-Spike protein levels. These biological findings do not show an abnormal frequency of ANA, RF, or anti-CCP and are consistent with the absence of autoimmune disease during the observation and follow-up period (4 years). This suggests a cellular rather than humoral inflammatory reaction [22], even if the anti-Spike protein antibodies induced by vaccination could mimic the ADE effect of antibodies produced during coronavirus infection. As with the response to corticosteroids in cases of ankle arthritis, treatment with low-dose corticosteroids is very often effective on other joint manifestations, but the lack of satisfactory response to corticosteroids seems to be related to the long persistence or increase of very high levels of anti-Spike (>2080 BAU/mL). Regardless, the significance of anti-Spike antibodies remains unclear; a high level of anti-Spike IGG antibodies was also detected in young patients who have developed post-vaccination myocarditis, especially in the presence of a high amount of Spike protein [23]. Even if the average age of late-onset patients is higher (51 years), we can also accredit the presence of high levels of anti-Spike IGG antibodies at the time of diagnosis and during the follow-up. As we do not clearly understand the mechanism controlling the quality and quantity of human protein production [24], let alone that of foreign viral protein, it is possible that the persistence of high levels of anti-Spike antibodies reflects the continued production of Spike protein. Although we cannot exclude the possibility that the onset of arthritis/polymyalgia after mRNA vaccination is a coincidence, the similarities between these two cohorts and the number of cases post vaccine over a short period of time might suggest some pathogenic causation. Normally, the level of post-vaccine-induced IGG anti-Spike lowered in 5 to 6 months, justifying the booster vaccine program [25], and after a 3rd dose of RNA vaccine, there is a reduction in anti-Spike levels from the 3rd month [26]. The presence of SARS-CoV-2 Spike antibodies more than a minimum of 8 months after a vaccine dose calls into question the physiopathology of early- and late-onset arthritis post-RNA vaccine. Vaccine dose count is higher in the late-onset ankle arthritis case series (47 injections) than in the early-onset group (25 injections). The comparison of early- and late-onset ankle arthritis reveals that the late-onset group received a higher number of DNA and RNA vaccine doses (8 DNA/39 RNA doses) compared to the early-onset case series (2 DNA/23 RNA doses). Additionally, the mean delay between the last vaccine dose and the rheumatological diagnosis is 17 months in the late-onset group, while it is only 0.44 months in the early-onset case series. The mechanism of COVID-19 vaccines is entirely different from that of live attenuated virus vaccines, thus rendering the analogical reasoning regarding the potential side effects incomplete. Post-COVID-19 vaccine late-onset adverse events prompt us to explore alternative pathways [6]. In vitro studies demonstrate the integration of RNA Spike sequences in human cells [27,28]. Spike protein could be the key to fixing the Angiotensin 2 receptors [6,29] and Toll-Like receptors 4 and 2 [6,30] implicated in an increased production of pro-inflammatory cytokines. Moreover, the strong affinity of Spike protein to Estrogen receptor alpha may explain why women are mostly involved. The number of injected DNA and RNA vaccine doses and the delay between the date of the last vaccine dose and the time of the rheumatological diagnosis might also argue that Spike protein could mediate early and delayed adverse events.

### Bradford Hill Criteria

This observational study satisfies all the Bradford Hill criteria. The high correlation coefficient observed in both cohorts and case series demonstrates the coherence and reproducibility of clinical and biologicals rheumatologic symptoms that manifest post-vaccination, indicating temporality. Most of the patients experienced adverse events following the 2nd or 3rd doses, and those with delayed onset ankle arthritis were vaccinated more frequently than those with early onset symptoms (dose effect). Recent works demonstrate the plausibility of the facts and the pro-inflammatory potential of Spike protein. Through the rheumatological manifestations observed, the study reveals the possibility that late side effects may appear after mRNA vaccination. As the mechanism of action of mRNA vaccines differs from that of attenuated or inactivated virus vaccines, other forms of side effects may emerge. Future controlled studies are needed to assess the long-term effects of anti-Spike protein RNA vaccines. Furthermore, to date, no patients with no previous history of rheumatological disease have developed autoimmune disease.

## 6. Conclusions

When encountering new cases of polymyalgia, arthritis, and particularly bilateral ankle arthritis in healthy individuals without any personal or family history of inflammatory diseases, it is essential to consider other potential causes of rheumatic or inflammatory diseases alongside post-vaccine adverse events. A strong positive result for plasmatic levels of anti-Spike protein suggests that these rheumatological manifestations might be triggered by inappropriate individual immune responses to the vaccine’s Spike protein and/or the overproduction of Spike protein, which can mediate a pro-inflammatory immune response explaining early- and late-onset effects.

## 7. Post Scriptum

When the observational study began on 13 September 2021, we found the subject without any reference points or published sources. This study must be viewed in the broader context of denying the existence of potential side effects. After one year, we were able to examine the time interval between the last vaccine dose and the measurement of anti-Spike antibody levels. At that time, based on references indicating a drop in post-vaccination Spike antibody levels within 3 to 4 months, public authorities estimated that a vaccine booster was necessary after 6 months. Given the known pathogenic role of the Spike protein, which justified the choice of RNA vaccination technology, we only measured anti-Spike IgG levels, which were paid for by the patients. Those who covered the costs of follow-up themselves reported any episodes of viral infection either spontaneously or during the guided medical history. Biological follow-ups were conducted, upon request, outside of any period of flu-like symptoms. Therefore, anti-N antibodies were not measured. Once again, as all intercurrent pathologies (e.g., one case of MGUS) were excluded, we found no hypergammaglobulinemia and no patients developed serum sickness or any autoimmune disease.

## 8. Illustrations



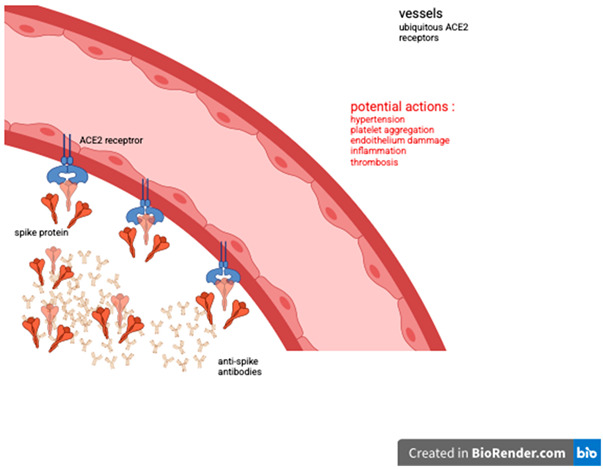





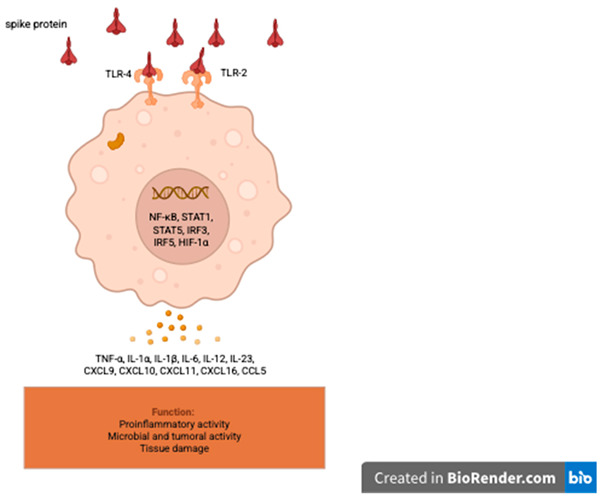





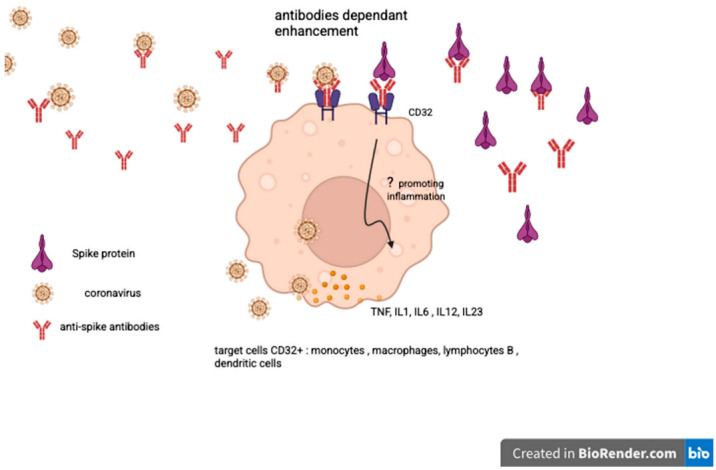



## Data Availability

Patients’ data are available only on request because of privacy and ethical restrictions.

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
