# Peer review of "Post-COVID-19 Vaccine Hyperproduction of Anti-Spike Antibodies and Rheumatological Manifestations"

_vaccines, 2025, doi:10.3390/vaccines13101028_

Round 1
Reviewer 1 Report
Comments and Suggestions for Authors
Vaccines-3881707:
HYPER PRODUCTION OF ANTI SPIKE ANTIBODIES AND RHEUMATOLOGICAL MANIFESTATIONS: COINCIDENCE OR PATHOGENESIS?, authored by Dr. Golstein Marc.
GENEAL COMMENTS:
This observational study investigated rheumatological manifestations following COVID-19 mRNA vaccination. As a results, most affected patients were women and the common symptoms included diffuse muscle pain resembling polymyalgia rheumatica and ankle arthritis accompanying high anti-Spike antibody levels. Strong correlations across cohorts and case series confirmed reproducible clinical and biological patterns. The authors concluded that the rheumatological symptoms may be triggered by inappropriate immune responses and/or overproduction of the Spike protein, leading to pro-inflammatory reactions that explain both early and late-onset adverse events. This is an interesting report in this field; however, there are several issues to be addressed.
SPECITFIC COMMENTS:
1. The manuscript would benefit from including a more detailed case report section. Presenting one or two representative cases with a detailed clinical course, medical history, medication use, and longitudinal data of polymyalgia rheumatica symptoms would provide valuable context.
2. While it is difficult to determine preexisting autoantibodies or rheumatic predisposition before vaccination, additional information on comorbidities such as sarcoidosis, autoimmune diseases, or relevant family history would strengthen the interpretation.
3. The predominance in women is consistent with long COVID, but could hormonal and generational factors also play a role? For example, estrogen decline in peri-/post-menopausal women or androgen decline in men of certain age groups might be relevant.
4. The HPA axis plays a key role in regulating immune responses. Please discuss whether any signs of ACTH or cortisol deficiency (preexisting or emerging after vaccination) could have contributed to the manifestations.
5. Since glucocorticoids are the mainstay of treatment for polymyalgia rheumatica and related rheumatological disorders, data on the patients' responsiveness to steroid therapy would be very informative.
6. If available, please provide laboratory data on immunological and inflammatory markers such as IgG4, IgE, CH50, ANCA, sIL2-R, or ACE. These markers may help in assessing potential autoimmune or granulomatous processes underlying vaccine-related reactions.
7. The authors mainly focus on anti-Spike antibody levels and the number of vaccine doses. However, it is important to consider additional factors such as the time interval between vaccination and antibody measurement, as well as the possibility of reinfection. Data on neutralizing antibodies, such as anti-N antibody titers, would also be valuable. Furthermore, information on total IgG levels and how these differed among individual patients could provide additional insights into the immunological background.
8. Finally, to aid broader readership, it would be useful to include a schematic figure illustrating how Spike protein and anti-Spike antibodies could potentially trigger rheumatological manifestations in this cohort.
9. As this manuscript uses the term pathogenesis in the title, I would note that there are varying scientific opinions regarding the causal role of COVID-19 vaccination. It may therefore be advisable to reconsider the title in light of the overall body of evidence.
Author Response
GENEAL COMMENTS:
This observational study investigated rheumatological manifestations following COVID-19 mRNA vaccination. As a results, most affected patients were women and the common symptoms included diffuse muscle pain resembling polymyalgia rheumatica and ankle arthritis accompanying high anti-Spike antibody levels. Strong correlations across cohorts and case series confirmed reproducible clinical and biological patterns. The authors concluded that the rheumatological symptoms may be triggered by inappropriate immune responses and/or overproduction of the Spike protein, leading to pro-inflammatory reactions that explain both early and late-onset adverse events. This is an interesting report in this field; however, there are several issues to be addressed.
SPECITFIC COMMENTS:
1. The manuscript would benefit from including a more detailed case report section. Presenting one or two representative cases with a detailed clinical course, medical history, medication use, and longitudinal data of polymyalgia rheumatica symptoms would provide valuable context.
229-277
2. While it is difficult to determine preexisting autoantibodies or rheumatic predisposition before vaccination, additional information on comorbidities such as sarcoidosis, autoimmune diseases, or relevant family history would strengthen the interpretation.
94-103
3. The predominance in women is consistent with long COVID, but could hormonal and generational factors also play a role? For example, estrogen decline in peri-/post-menopausal women or androgen decline in men of certain age groups might be relevant.
304-307
310-314
4. The HPA axis plays a key role in regulating immune responses. Please discuss whether any signs of ACTH or cortisol deficiency (preexisting or emerging after vaccination) could have contributed to the manifestations.
94-103
5. Since glucocorticoids are the mainstay of treatment for polymyalgia rheumatica and related rheumatological disorders, data on the patients' responsiveness to steroid therapy would be very informative.
322-323
327-327
6. If available, please provide laboratory data on immunological and inflammatory markers such as IgG4, IgE, CH50, ANCA, sIL2-R, or ACE. These markers may help in assessing potential autoimmune or granulomatous processes underlying vaccine-related reactions.
94-103
112-114
149-150
176-178
7. The authors mainly focus on anti-Spike antibody levels and the number of vaccine doses. However, it is important to consider additional factors such as the time interval between vaccination and antibody measurement, as well as the possibility of reinfection. Data on neutralizing antibodies, such as anti-N antibody titers, would also be valuable. Furthermore, information on total IgG levels and how these differed among individual patients could provide additional insights into the immunological background.
281-302
416-433
8. Finally, to aid broader readership, it would be useful to include a schematic figure illustrating how Spike protein and anti-Spike antibodies could potentially trigger rheumatological manifestations in this cohort.
see illustrations 510-527
9. As this manuscript uses the term pathogenesis in the title, I would note that there are varying scientific opinions regarding the causal role of COVID-19 vaccination. It may therefore be advisable to reconsider the title in light of the overall body of evidence.
1-2
Reviewer 2 Report
Comments and Suggestions for Authors
The authors present a study in which they wonder whether the concentration of SARS-CoV-2 anti-spike antibodies after vaccination coincides with the onset of rheumatologic disease.
Comments:
- It should be clear from the title that the presence of anti-spike antibodies has been studied in relation to vaccination.
- The abstract should be better structured, including background, objective, methods, results, and conclusion. In addition, it should be consistent with regard to verb tense, which should be in the past rather than in the present.
- The vaccines ChAdOx1 and CoV-19 are not properly DNA vaccines but viral vector vaccines
- The introduction is rather limited and should be sufficiently expanded with an explanation of the issues related to vaccination and with relevant data that may correlate it with the onset of autoimmune rheumatic diseases.
- The tables are not sufficiently clear for the interpretation of the results. It might be preferable to present part of the results in figures, for instance using bar charts, in order to provide a more immediate perception of the correlation between the different rheumatic diseases and antibody titers.
- Instead of using the verb 'to monster', it would be better to choose a more appropriate term, such as 'to show' (line 203).
- The authors should conduct an analysis to determine the odds ratio of developing an autoimmune rheumatic disease as a function of high antibody titers.
- The authors should clarify whether, in the presence of high titers of antibodies, these autoimmune conditions tend to resolve over time or whether the presence of elevated anti-spike antibody titers may contribute to the development of chronic autoimmune rheumatic disease.
- It would also be useful to add a figure illustrating the mechanism through which anti-spike antibodies may induce an inflammatory disease.
- The authors should also comment on whether there are data in the literature regarding the onset of rheumatic diseases after natural SARS-CoV-2 infection, and whether studies have been conducted on possible correlations between such events and the antibody titers developed.
The English language should be improved, as in some parts the text is difficult to understand
Author Response
- It should be clear from the title that the presence of anti-spike antibodies has been studied in relation to vaccination.
- 1-2
- The abstract should be better structured, including background, objective, methods, results, and conclusion. In addition, it should be consistent with regard to verb tense, which should be in the past rather than in the present.
- 39-68
- The vaccines ChAdOx1 and CoV-19 are not properly DNA vaccines but viral vector vaccines
- Done in the text ( many thanks )
- The introduction is rather limited and should be sufficiently expanded with an explanation of the issues related to vaccination and with relevant data that may correlate it with the onset of autoimmune rheumatic diseases.
- yes see modifications 71-86
- The tables are not sufficiently clear for the interpretation of the results. It might be preferable to present part of the results in figures, for instance using bar charts, in order to provide a more immediate perception of the correlation between the different rheumatic diseases and antibody titers.
- table 2-3-4 modified
- Instead of using the verb 'to monster', it would be better to choose a more appropriate term, such as 'to show' (line 203).
- done
- The authors should conduct an analysis to determine the odds ratio of developing an autoimmune rheumatic disease as a function of high antibody titers.
- we didn't diagnose any auto immune disease during the follow-up
- 95-103, 149-150, 177-178, 416-433
- The authors should clarify whether, in the presence of high titers of antibodies, these autoimmune conditions tend to resolve over time or whether the presence of elevated anti-spike antibody titers may contribute to the development of chronic autoimmune rheumatic disease.
- 344-346
- It would also be useful to add a figure illustrating the mechanism through which anti-spike antibodies may induce an inflammatory disease.
- yes , see illustrations 510-529
- The authors should also comment on whether there are data in the literature regarding the onset of rheumatic diseases after natural SARS-CoV-2 infection, and whether studies have been conducted on possible correlations between such events and the antibody titers developed.
- 291-301
All of the reviewers' comments were invaluable to me, not only in significantly improving the text, but also in deepening my knowledge.
Round 2
Reviewer 1 Report
Comments and Suggestions for Authors
The authors modified their manuscript and added new data, however, the responses to the referees were not appropriate in the response sheets. The authors need to address how and what points were amended or modified based on the referees' comments.
Author Response
GENEAL COMMENTS:
This observational study investigated rheumatological manifestations following COVID-19 mRNA vaccination. As a results, most affected patients were women and the common symptoms included diffuse muscle pain resembling polymyalgia rheumatica and ankle arthritis accompanying high anti-Spike antibody levels. Strong correlations across cohorts and case series confirmed reproducible clinical and biological patterns. The authors concluded that the rheumatological symptoms may be triggered by inappropriate immune responses and/or overproduction of the Spike protein, leading to pro-inflammatory reactions that explain both early and late-onset adverse events. This is an interesting report in this field; however, there are several issues to be addressed.
SPECITFIC COMMENTS:
1. The manuscript would benefit from including a more detailed case report section. Presenting one or two representative cases with a detailed clinical course, medical history, medication use, and longitudinal data of polymyalgia rheumatica symptoms would provide valuable context.
Please find here under 3 cases
229-277
Example of longitudinal case studies:
Case 1: A 71-year-old Caucasian woman with a medical history of hypertension and hypercholesterolemia consulted for bilateral ankle synovitis that appeared 8 months ago, 15 days after receiving the third dose of the Pfizer vaccine. At the time of consultation, blood tests revealed a CRP level of 2 mg/L, no autoantibodies, and anti-Spike antibodies > 2080 BAU/mL (8 months post-vaccine). Comprehensive investigations for bilateral arthritis showed no abnormalities. The patient improved with 8 mg/day of methylprednisolone and was instructed to reduce the dose to 4 mg/day after 15 days. After1 month, a follow-up blood test showed an SR of 4 mm/h, CRP of 2 mg/L, and anti-Spike antibodies > 2080 BAU/mL (9 months post-vaccine). The synovitis had disappeared, and the methylprednisolone dose was reduced to 2 mg/day. Four months later, at a follow-up consultation, the SR was 10 mm/h, CRP was 1 mg/L, and anti-Spike antibodies remained > 2080 BAU/mL (13 months post-vaccine). The patient requested to stop the corticosteroid treatment. 3 Months later, the patient consulted again due to the reappearance of bilateral ankle synovitis. Usual tests did not show elevated SR (6 mm/h) or CRP (2 mg/L), but anti-Spike antibodies remained > 2080 BAU/mL (16 months post-vaccine). The patient improved with 4 mg of methylprednisolone. Once again, she stopped the treatment and later consulted for bilateral ankle arthritis. Blood test results showed an SR of 4 mm/h, CRP of 20 mg/L, and anti-Spike antibodies > 2080 BAU/mL (20 months post-vaccine). The patient recovered rapidly with a daily dose of 4 mg of methylprednisolone. This time, she maintained the treatment, and follow-up tests showed normal SR (4 mm/h) and CRP (1 mg/L), but anti-Spike antibodies remained very high (> 2080 BAU/ml 25 months post-vaccine).
Case 2: A 56-year-old Caucasian woman with no medical history consulted for bilateral shoulder pain and polymyalgia that appeared 17 months ago 2 months after receiving the second dose of the Pfizer vaccine. At the time of consultation, blood tests revealed an SR level of 26 mm/h, a CRP level of 27 mg/L, no autoantibodies, and anti-Spike antibodies at 1070 BAU/mL (17 months post-vaccine). Investigations for bilateral arthritis showed no abnormalities. The patient improved with 8 mg/day of methylprednisolone and was instructed to reduce the dose to 4 mg/day after 15 days. She stopped the treatment spontaneously and later consulted for polymyalgia. Blood test results showed an SR of 38 mm/h, CRP of 32 mg/L, and anti-Spike antibodies > 2080 BAU/mL (22 months post-vaccine). The patient recovered rapidly with a daily dose of 4 mg of methylprednisolone. She maintained the treatment, and follow-up tests showed an SR of 29 mm/h, CRP of 33 mg/L, and anti-Spike antibodies remained high at 1160 BAU/ml (31 months post-vaccine).
Case 3: A 55-year-old Caucasian man with a medical history of cervical osteoarthritis consulted for bilateral ankle synovitis and polymyalgia that appeared 8 months ago, more than 3 months after receiving the fourth dose of the Pfizer vaccine. At the time of consultation, blood tests revealed an ESR of 33 mm/h, a CRP level of 27 mg/L, no autoantibodies, and anti-Spike antibodies > 2080 BAU/mL (8 months after the last vaccine). Tests for bilateral arthritis revealed no abnormalities. The patient's condition improved with 8 mg/day of methylprednisolone and was instructed to reduce the dose to 4 mg/day after 15 days. Blood test results showed an ESR of 2 mm/h, CRP of 2 mg/L, and anti-Spike antibodies > 2030 BAU/mL (13 months after vaccination). He continued treatment and follow-up tests showed an ESR of 2 mm/h, a CRP of 1 mg/L and anti-Spike antibodies still elevated at 1270 BAU/mL (17 months after vaccination). Corticosteroid treatment was discontinued.
2. While it is difficult to determine preexisting autoantibodies or rheumatic predisposition before vaccination, additional information on comorbidities such as sarcoidosis, autoimmune diseases, or relevant family history would strengthen the interpretation.
94-104
We exclude of the studies all patients having a past of rheumatic, inflammatory, granulomatous or auto immune diseases. As we also excluded from our study anyone with anxiety depression, fibromyalgia or any other psychiatric condition, we did not prospectively study the HPA ACTH cortisol axis. Furthermore, none of the patients included presented with complaints or clinical symptoms related to abnormal ACTH-cortisol activity. Specific tests such as CH50, C3, C4, ACE, or immunoglobulin typing were not systematically requested at the start of the study. These were performed on an exceptional basis depending on clinical and/or biological findings. No abnormalities were found except for elevated ACE in the case of sarcoidosis (excluded of the study) that appeared 3 months after RNA vaccination
Using ABRUMET ( general medical data of each patient) , we confirm the absence of pre existing rheumatic or inflammatory diseases and during the follow up of both cohorts
3. The predominance in women is consistent with long COVID, but could hormonal and generational factors also play a role? For example, estrogen decline in peri-/post-menopausal women or androgen decline in men of certain age groups might be relevant.
I cannot fully respond to this comment because we did not ask all patients about possible menstrual disorders in non-menopausal women, so we are unable to properly compare non-menopausal women with those who are menopausal. This would indeed be a very interesting area of research.
305-308
We also chose to exclude patients with anxiety depression, fibromyalgia, chronic fatigue syndrome or other psychiatric conditions to avoid any diagnosis of long Covid [11], whose symptoms are essentially subjective and overlap with those of other conditions.
310-315
Estrogens have a favorable anti-inflammatory and immunomodulatory effect, providing protection during the SARS-CoV-2 epidemic [12]. The selective affinity of the spike protein for estrogen alpha receptors [5] could also explain the predominance of biological inflammatory syndrome in women, as it competes with endogenous or replacement estradiol for binding to its receptor during menopause.
4. The HPA axis plays a key role in regulating immune responses. Please discuss whether any signs of ACTH or cortisol deficiency (preexisting or emerging after vaccination) could have contributed to the manifestations.
Without any literature data on this specific post vaccine subject in 2021 and clinical signs, we did not measure HPA axis, ACTH or cortisol levels.
97-104
As we also excluded from our study anyone with anxiety depression, fibromyalgia or any other psychiatric condition, we did not prospectively study the HPA ACTH cortisol axis. Furthermore, none of the patients included presented with complaints or clinical symptoms related to abnormal ACTH-cortisol activity. Specific tests such as CH50, C3, C4, ACE, or immunoglobulin typing were not systematically requested at the start of the study. These were performed on an exceptional basis depending on clinical and/or biological findings. No abnormalities were found except for elevated ACE in the case of sarcoidosis (excluded of the study) that appeared 3 months after RNA vaccination
5. Since glucocorticoids are the mainstay of treatment for polymyalgia rheumatica and related rheumatological disorders, data on the patients' responsiveness to steroid therapy would be very informative.
321-324
After excluding most causes of diffuse pain, 8 mg of methylprednisolone per day, reduced to 4 mg per day, was administered for other clinical manifestations and proved effective in most cases (65 out of 73 in cohort 1 and 35 out of 53 in cohort 2).
326-328
In our initial study of 17 cases of early-onset bilateral ankle arthritis [16], 16 out of 17 patients responded favorably to low-dose corticosteroid therapy, and 13 out of 17 in the series of late-onset ankle arthritis
6. If available, please provide laboratory data on immunological and inflammatory markers such as IgG4, IgE, CH50, ANCA, sIL2-R, or ACE. These markers may help in assessing potential autoimmune or granulomatous processes underlying vaccine-related reactions.
ANCA levels were measured in all patients in the first cohort and were not found in any patients. Their measurement was not performed subsequently. Sil2-R levels were not measured. IGG, IGE, CH50 and ACE typing were only requested on a case-by-case basis depending on the clinical presentation.
100-104
Specific tests such as CH50, C3, C4, ACE, or immunoglobulin typing were not systematically requested at the start of the study. These were performed on an exceptional basis depending on clinical and/or biological findings. No abnormalities were found except for elevated ACE in the case of sarcoidosis (excluded of the study) that appeared 3 months after RNA vaccination.
113-115
We also focused our biological investigation on anti-nuclear antibodies (ANA), rheumatoid factor (RF), anti-citrullinated peptide (anti-CCP) and anti-neutrophil cytoplasmic antibodies (ANCA).
150-151
ANCA were not detected in any patients and in the absence of signs of vasculitis or autoimmune disease
177-179
As ANCA were not detected in the first cohort, we did not measure them again. No abnormalities in CH50 C3 and C4 testing’s were found in a patient presenting with livedo reticularis.
7. The authors mainly focus on anti-Spike antibody levels and the number of vaccine doses. However, it is important to consider additional factors such as the time interval between vaccination and antibody measurement, as well as the possibility of reinfection. Data on neutralizing antibodies, such as anti-N antibody titers, would also be valuable. Furthermore, information on total IgG levels and how these differed among individual patients could provide additional insights into the immunological background.
296-303
The choice of anti-Spike antibody dosage was motivated by the mode of action of anti-Covid-19 vaccines, which focus on the Spike protein [1,2]. On the other hand, during coronavirus infections, anti-Spike antibodies have an antibody-dependent enhancement (ADE)effect, promoting viral infection and stimulating certain pro-inflammatory cytokines [9,10]. This also raises questions about the possible role of hyperproduction of anti-Spike protein antibodies in the emergence of rheumatological manifestations by fixing viral particle (Spike protein) and mimicking the ADE mechanism.
When the observational study began on September 13, 2021, we found the subject without any reference points or published sources. This study must be viewed in the broader context of denying the existence of potential side effects. After one year, we were able to examine the time interval between the last vaccine dose and the measurement of anti-spike antibody levels. At that time, based on references indicating a drop in post-vaccination spike antibody levels within 3 to 4 months, public authorities estimated that a vaccine booster was necessary after 6 months. Given the known pathogenic role of the spike protein, which justified the choice of RNA vaccination technology, we only measured anti-spike IgG levels, which were paid for by the patients. Those who covered the costs of follow-up themselves reported any episodes of viral infection either spontaneously or during the guided medical history. Biological follow-ups were conducted, upon request, outside of any period of flu-like symptoms. Therefore, anti-N antibodies were not measured. Once again, as all intercurrent pathologies (e.g., 1 case of MGUS) were excluded, we found no hypergammaglobulinemia and no patients developed serum sickness or any autoimmune disease.
8. Finally, to aid broader readership, it would be useful to include a schematic figure illustrating how Spike protein and anti-Spike antibodies could potentially trigger rheumatological manifestations in this cohort.
see illustrations 510-527
9. As this manuscript uses the term pathogenesis in the title, I would note that there are varying scientific opinions regarding the causal role of COVID-19 vaccination. It may therefore be advisable to reconsider the title in light of the overall body of evidence.
the title is changed
I would like to take this opportunity to express my sincere gratitude to the reviewers for their comments, which have greatly improved my text and my own knowledge.
Reviewer 2 Report
Comments and Suggestions for Authors
The manuscript has been extensively revised, and responses have been provided to all of the reviewer’s comments.
Author Response
- It should be clear from the title that the presence of anti-spike antibodies has been studied in relation to vaccination.
- yes the titel was changed
- The abstract should be better structured, including background, objective, methods, results, and conclusion. In addition, it should be consistent with regard to verb tense, which should be in the past rather than in the present.
- 39-68
-
Abstract
Introduction
Vaccines are the most widely used public health measure to control the global COVID-19 pandemic. Most vaccines used in Europe and North America are mRNA-based. A mass vaccination campaign was carried out between 2021 and 2024. Some adverse events have been reported based on analogies with previous virus-attenuated vaccines.
Objectives
Given the new mechanism inducing specific antibodies, we questioned the role of mRNA Spike vaccines and the significance of hyperproduction of anti-Spike antibodies in the emergence of early and late onset rheumatological manifestations observed after one or more injections.
Material and Methods
A prospective observational study involving two cohorts was initiated. The first cohort was observed from September 13, 2021, to September 30, 2022, and the second cohort from October 1, 2022, to September 30, 2023. The study also focused on the interval between the last vaccine injection and the onset of rheumatic symptoms. None of the patients had a history of rheumatic or inflammatory diseases. We compared both cohorts and ankle arthritis case series to analyze the differences between early and late-onset adverse events.
Results
In both cohorts and case series, the majority of patients were women. The most common symptoms included diffuse muscle pain, which mimics polymyalgia rheumatica, and ankle arthritis. Very high levels of anti-Spike antibodies (> 2080 BAU/ml) were generally detected. The Pearson correlation coefficient between both cohorts and case series was very high, confirming the reproducibility of post-vaccine clinical and biological features.
Conclusion
These rheumatological manifestations might be triggered by inappropriate individual immune responses to the vaccine's Spike protein and/or the overproduction of Spike protein, which can mediate a pro-inflammatory reaction explaining early and late-onset effects.
- The vaccines ChAdOx1 and CoV-19 are not properly DNA vaccines but viral vector vaccines
- Done in the text ( many thanks )
- The introduction is rather limited and should be sufficiently expanded with an explanation of the issues related to vaccination and with relevant data that may correlate it with the onset of autoimmune rheumatic diseases.
- yes see modifications 71-86
-
Introduction:
Vaccination is the most widely used public health measure to control the global COVID-19 pandemic. Most vaccines used in Europe and North America are mRNA-based, such as BNT162b2 and mRNA-1273, but viral vector DNA vaccines like ChadOx1 and CoV-19 are also used [1,2]. A mass vaccination campaign was launched between 2021 and 2024. Some adverse events have been reported, drawing analogies from side effects induced by attenuated or inactivated virus vaccines. A few rheumatological events have been documented [3,4], occurring rapidly after the administration of one or more doses. The spike protein targeted by the humoral immune response binds to the ACE2 receptor, blocking SARS-CoV-2 entry and preventing inflammation. However, the spike protein also binds to the alpha estrogenic receptor and the Toll 2 and 4 receptors [5,6]. Furthermore, the feedback mechanisms governing the quantity and quality of foreign protein production by the endoplasmic reticulum and Golgi apparatus in humans remain unknown [7]. Due to the different and specific mechanisms of Spike protein RNA vaccines or viral vector DNA vaccines, we question the role of COVID-19 vaccines in the emergence of early and late-onset rheumatological manifestations observed following one or more injections.
- The tables are not sufficiently clear for the interpretation of the results. It might be preferable to present part of the results in figures, for instance using bar charts, in order to provide a more immediate perception of the correlation between the different rheumatic diseases and antibody titers.
- table 2-3-4 modified
- Instead of using the verb 'to monster', it would be better to choose a more appropriate term, such as 'to show' (line 203).
- done
- The authors should conduct an analysis to determine the odds ratio of developing an autoimmune rheumatic disease as a function of high antibody titers.
- we didn't diagnose any auto immune disease during the follow-up
- 95-103, 149-150, 177-178, 416-433
- The authors should clarify whether, in the presence of high titers of antibodies, these autoimmune conditions tend to resolve over time or whether the presence of elevated anti-spike antibody titers may contribute to the development of chronic autoimmune rheumatic disease.
- 345-347
- These biological findings do not show an abnormal frequency of ANA, RF, or anti-CCP and are consistent with the absence of autoimmune disease during the observation and follow-up period (4 years).
- It would also be useful to add a figure illustrating the mechanism through which anti-spike antibodies may induce an inflammatory disease.
- yes , see illustrations 1-2 -3
- The authors should also comment on whether there are data in the literature regarding the onset of rheumatic diseases after natural SARS-CoV-2 infection, and whether studies have been conducted on possible correlations between such events and the antibody titers developed.
- 292-303
- A few cases of reactive arthritis following Covid infection have been reported. The incidence appears to be very low and inversely proportional to the severity of the disease [8]. As only vaccinated patients were included in the study, the vast majority of whom had received two or three doses, the risk of post-infectious rheumatological or inflammatory manifestations is very limited. The choice of anti-Spike antibody dosage was motivated by the mode of action of anti-Covid-19 vaccines, which focus on the Spike protein [1,2]. On the other hand, during coronavirus infections, anti-Spike antibodies have an antibody-dependent enhancement (ADE)effect, promoting viral infection and stimulating certain pro-inflammatory cytokines [9,10]. This also raises questions about the possible role of hyperproduction of anti-Spike protein antibodies in the emergence of rheumatological manifestations by fixing viral particle (Spike protein) and mimicking the ADE mechanism.
Round 3
Reviewer 1 Report
Comments and Suggestions for Authors
The authors revised their manuscript based on the referees' comments.